# Mitochondrial Oxidative Stress Induces Cardiac Fibrosis in Obese Rats through Modulation of Transthyretin

**DOI:** 10.3390/ijms23158080

**Published:** 2022-07-22

**Authors:** Ernesto Martínez-Martínez, Joaquín Fernández-Irigoyen, Enrique Santamaría, María Luisa Nieto, José Manuel Bravo-San Pedro, Victoria Cachofeiro

**Affiliations:** 1Departamento de Fisiología, Facultad de Medicina, Universidad Complutense de Madrid, 28040 Madrid, Spain; josemabr@ucm.es; 2Ciber de Enfermedades Cardiovasculares (CIBERCV), Instituto de Salud Carlos III, 28222 Madrid, Spain; mlnieto@ibgm.uva.es; 3Instituto de Investigación Sanitaria Gregorio Marañón (IiSGM), 28007 Madrid, Spain; 4Proteomics Platform, Navarrabiomed, Hospital Universitario de Navarra (HUN), Universidad Pública de Navarra (UPNA), IdiSNA, 31008 Pamplona, Spain; jfernani@navarra.es (J.F.-I.); enrique.santamaria.martinez@navarra.es (E.S.); 5Instituto de Biología y Genética Molecular, CSIC-Universidad de Valladolid, 47002 Valladolid, Spain

**Keywords:** endoplasmic reticulum stress, fibrosis, mitochondrial oxidative stress, obesity, transthyretin

## Abstract

A proteomic approach was used to characterize potential mediators involved in the improvement in cardiac fibrosis observed with the administration of the mitochondrial antioxidant MitoQ in obese rats. Male Wistar rats were fed a standard diet (3.5% fat; CT) or a high-fat diet (35% fat; HFD) and treated with vehicle or MitoQ (200 μM) in drinking water for 7 weeks. Obesity modulated the expression of 33 proteins as compared with controls of the more than 1000 proteins identified. These include proteins related to endoplasmic reticulum (ER) stress and oxidative stress. Proteomic analyses revealed that HFD animals presented with an increase in cardiac transthyretin (TTR) protein levels, an effect that was prevented by MitoQ treatment in obese animals. This was confirmed by plasma levels, which were associated with those of cardiac levels of both binding immunoglobulin protein (BiP), a marker of ER stress, and fibrosis. TTR stimulated collagen I production and BiP in cardiac fibroblasts. This upregulation was prevented by the presence of MitoQ. In summary, the results suggest a role of TTR in cardiac fibrosis development associated with obesity and the beneficial effects of treatment with mitochondrial antioxidants.

## 1. Introduction

Obesity is defined as abnormal or excessive fat accumulation and is considered a major risk to health. A body mass index over 30 kg/m^2^ is considered to be obese. The prevalence of obesity has increased drastically in recent years not only in adults but also in children, manifesting itself as a global health problem. Obesity has a considerable impact on metabolism, affecting the entire organism [1] and predisposing to comorbidities such as cancer [2], hypertension [3], insulin resistance and diabetes [4], neurological disorders [5] and cardiovascular diseases, including heart failure [1]. At the cardiac level, obesity exponentially increases the risk of heart failure development due to the impact of obesity on the heart [6]. An excessive body weight generates higher metabolic demands with consequent increases in cardiac output and workload [7]. Obesity has been described to promote left ventricular hypertrophy and diastolic dysfunction, in which the development of cardiac fibrosis plays a central role [8].

Fibrosis is defined as excessive extracellular matrix accumulation, which could be due to an increase in extracellular matrix protein synthesis and/or a decrease in its degradation. It is a well-recognized cause of high morbidity and mortality, increasing myocardial stiffness and, therefore, promotion of diastolic dysfunction [9,10]. Furthermore, its evaluation has been suggested as a useful indicator of long-term mortality in heart failure patients [11]. Different treatments have been recommended by the official guidelines to improve clinical symptoms in heart failure patients, showing promising results; however, it is important to mention that these strategies are not effective in reversing cardiac fibrosis [12]. Therefore, it is important to understand the mechanisms involved in cardiac fibrosis development and to design new, effective pharmacological treatments capable of preventing the cardiac consequences of fibrosis development.

Several mechanisms have been proposed to be involved in cardiac fibrosis in the context of obesity, such as neurohormonal alterations [13], inflammation [14], hyperleptinemia [15], oxidative stress [16] and endoplasmic reticulum (ER) stress [16], among others. ER stress is a situation characterized by an increase in unfolded/misfolded proteins in the lumen of the ER. Under these conditions, binding immunoglobulin protein (BiP) is dissociated from three transmembrane proteins presented in the ER, such as inositol-requiring 1, PKR-like ER kinase and activating transcription factor 6 (ATF6), thereby activating different downstream pathways in order to restore ER homeostasis [17]. If the ER stress is maintained, the ER leaks calcium, which could damage the mitochondria, thereby generating an excess of free radicals and promoting the activation of apoptosis, in turn generating cell death and tissue injury [18]. In previous studies from our group, we evaluated the role of mitochondrial oxidative stress in some of these mechanisms involved in cardiac alterations associated with obesity. In this sense, treatment for 6 weeks with the mitochondrial antioxidant MitoQ prevented cardiac lipotoxicity characterized by mitochondrial lipid remodeling [16]. In addition, we observed that treatment with MitoQ prevented the development of cardiac fibrosis and the increase in collagen I protein levels, as well as the activation of ER stress in obese animals, showing an interaction between mitochondrial oxidative stress and ER stress and its involvement in cardiac fibrosis observed in obese rats [19]. In a previous study from our group, obese animals presented with an increase in BiP protein levels accompanied by an upregulation of CCAAT-enhancer-binding homologous protein (CHOP), a downstream protein induced by ER stress activation [19]. In addition, in cardiac cells, we confirmed direct crosstalk between ER stress and mitochondrial oxidative stress and its association with extracellular matrix protein synthesis [19]. However, the entire effect of obesity on cardiac alterations and the involvement of mitochondrial oxidative stress have not been fully established. The aim of this study was to delve into the benefits observed in obese animals treated with the mitochondrial antioxidant MitoQ. A proteomic approach was used for the characterization of proteostasis improvement after MitoQ treatment in order to establish the protein interactome network regulated by obesity and mitochondrial oxidative stress.

## 2. Results

### 2.1. Proteome-Wide Exploration of Obesity at the Cardiac Level

In a previous study, we demonstrated that animals fed a high-fat diet for 6 weeks showed an increase in body weight as compared with controls. The administration of MitoQ reduced this increase; however, animals subjected to such administration still had higher body weights than those in the control group [20]. These changes in body weight could be, at least in part, due to differences in energy intake (calculated from the diet-contained calories) observed in the different groups [20]. Obese rats presented with normal cardiac structure, systolic and diastolic cardiac function as evaluated by left ventricle ejection fraction and E/A ratio, as well as blood pressure levels. However, despite no observed changes in cardiac function due to the short time of evolution with the diet, obese animals presented with important cardiac alterations characterized by interstitial fibrosis and an increase in relative heart weight and cardiomyocyte area accompanied by an increase in oxidative stress and activation in ER stress [16,19]. Treatment with the mitochondrial antioxidant MitoQ prevented all of these alterations. This beneficial effect of MitoQ seems to occur through, at least in part, the interaction between oxidative stress and ER stress in extracellular matrix production [16,19].

In order to obtain a deep insight into the cardiac protein content and protein function modulated by obesity and MitoQ treatment, a proteome-wide analysis of total cardiac extracts was performed using 2D nano-liquid chromatography tandem mass spectrometry. More than 1000 proteins were identified and quantified in the proteomic analysis. HFD modulated the expression of 33 proteins as compared with a control diet (CT) (Appendix A), including proteins related to ER stress, lipid metabolism, oxidative stress and the immune system. Obese animals treated with the mitochondrial antioxidant presented with 23 altered protein expressions as compared with untreated obese animals (Appendix A).

Interestingly, among the modulated proteins observed in the proteomic analysis, an increase in transthyretin (TTR) and 14:3:3 protein eta at the cardiac level was observed in obese animals as compared to control animals. Treatment with the mitochondrial antioxidant reverted these changes observed in obese animals, showing differential protein expression under the experimental conditions (Figure 1A,B). The changes observed in TTR levels were confirmed at plasma levels, and treatment with MitoQ prevented the increase in TTR circulating levels observed in obese animals (Figure 1C).

### 2.2. Protein Interactome Network Modulated by TTR

To enhance the analytical outcome of proteomic findings, TTR protein interactome and multipathway analyses were generated. The most significantly disrupted biofunctions were response to topologically incorrect protein (LogP = −9.52), regulation of protein stability (LogP = −8.75), cellular response to stimuli (LogP = −7.31), regulation of cellular localization (LogP = −6.86), cellular response to peptides (LogP = −6.68), insertion of tail-anchored proteins into the ER membrane (LogP = −6.38) and response to misfolded protein (LogP = −6.22) (Figure 2A and Appendix A associated with Figure 2A). In addition, we used Biogrid to define potential regulators involved in the cardiac alterations observed in obese animals. According to information present in the Biogrid database, we observed that ATF4 (activating transcription factor 4), a protein involved in ER stress, was experimentally demonstrated as a TTR protein interactor (Figure 2B).

Interestingly, direct correlations were found between TTR circulating levels and myocardial BiP protein levels, a marker of ER stress activation (*p* < 0.0016; Figure 3A). In addition, TTR was correlated with cardiac fibrosis (*p* < 0.0112; Figure 3B) and collagen type I protein levels (*p* < 0.0418; Figure 3C). These results suggest the following: (i) the possible participation of TTR in the cardiac fibrosis observed in obese animals and (ii) an additional mechanism for the beneficial effects of treatment with the mitochondrial antioxidant at the cardiac level in obese animals.

### 2.3. Effects of TTR on Cardiac Fibroblasts

In order to analyze the direct effects of TTR on extracellular matrix proteins and ER stress activation, cardiac fibroblasts were exposed to recombinant TTR. Cardiac fibroblasts treated with TTR presented with an increase in collagen type I protein levels, reaching a maximum effect in TTR-treated cells at 5 µg/mL (Figure 4A,B).

Recombinant TTR increased BiP protein levels in a dose-dependent manner in cardiac fibroblasts, indicating activation of ER stress (Figure 4A,C). Analysis of different pathways involved in ER stress activation revealed that cardiac cells treated with TTR presented with enhanced protein levels of CHOP and ATF6⍺ levels in a dose-dependent manner (Figure 4A,D,E). Original blots are presented in Appendix A.

### 2.4. Mitochondrial Oxidative Stress Mediates the Effects of TTR on Cardiac Fibroblasts

Once the direct effects of TTR on extracellular matrix production and ER stress were confirmed, cardiac fibroblasts were exposed to TTR (5 µg/mL) in the presence or absence of the mitochondrial antioxidant MitoQ (5 nM).

The presence of MitoQ in the cultured medium blunted the profibrotic effect of TTR, preventing an increase in collagen type I protein levels induced by TTR (Figure 5A,B). This protective effect of MitoQ could be, at least in part, due to the prevention of ER stress activation induced by TTR. As previously described, TTR increased BiP protein levels, an effect that was accompanied by an upregulation of CHOP in the absence of modifications in ATF6⍺ protein levels after 24 h of stimulation (Figure 5A,C–E). The presence of MitoQ in the culture medium prevented the activation of ER stress induced by TTR (Figure 5A,C–E). Original blots are presented in Appendix A.

## 3. Discussion

The novel findings of the present study are as follows: (1) TTR is increased at cardiac and plasma levels in obese animals, even in the absence of functional cardiac alterations; (2) mitochondrial oxidative stress promotes TTR upregulation in the heart of obese rats; and (3) TTR exerts profibrotic actions and ER stress activation in cardiac cells through mitochondrial oxidative stress. These findings elucidate novel mechanisms of cardiac alterations associated with obesity and suggest a possible new approach to TTR-related pathologies, such as cardiac amyloidosis.

Obesity is associated with several cardiac alterations, such as structural modifications, hypertrophy and fibrosis, which ultimately promotes functional alterations. In the present study, HFD animals presented with cardiac fibrosis and hypertrophy characterized by an increase in relative heart weight and cardiomyocyte cross-sectional area. These alterations occurred prior to functional modifications, as obese animals showed normal systolic and diastolic function as evaluated by left ventricle ejection fraction and E/A ratio [16], respectively. In addition, obese rats did not develop diabetes but presented with insulin resistance [20] due to the short time of HFD. Previous studies have demonstrated that long-term feeding with HFD or gene mutations promote obesity, diabetes and cardiac functional alterations in animals, as well as cardiac hypertrophy and fibrosis [21,22,23]. The purpose of the present study was to evaluate the effects of the mitochondrial antioxidant MitoQ in an early stage of cardiac damage in an animal model of obesity.

Proteomic analyses identified differential expression of two proteins at the cardiac level: TTR and 14-3-3ζ. TTR, also known as prealbumin, is a 55 kDa tetrameric protein mainly synthesized in the liver; its main function is to act as a plasma carrier protein of thyroxin and retinol [24,25]. In a previous study, these two proteins were found to be interconnected, thereby showing that TTR regulates 14-3-3ζ protein levels in the hippocampus of young mice [26]. In obese patients, an increase was observed in TTR expression, suggesting it as a biomarker for diabetes progression in overweight patients [27]. In accordance with these findings, we observed an increase in TTR plasma levels in obese rats with insulin resistance, as suggested by an increase in HOMA index values [20]. Previous studies have demonstrated that TTR is related to oxidative stress, as its levels correlate with reactive oxygen and nitrogen species [28,29], in addition to inducing oxidative stress [25]. In the present study, the increase in TTR protein levels observed in obese animals was accompanied by an increase in superoxide anion cardiac levels [16]. Interestingly, treatment with the mitochondrial antioxidant MitoQ prevented an increase in TTR plasma levels, as well as an increase in HOMA index values in HFD rats. It is important to mention that treatment with MitoQ reduced but did not normalize the increase in body weight induced by an HFD. These results confirm the interaction between TTR and insulin resistance and show, for the first time, that TTR is not only a biomarker of oxidative stress but also an oxidative environment, which promotes an increase in TTR levels.

TTR is also related to other pathologies due to its misfolding or unfolding properties, promoting aggregates and amyloid fibrils and thereby contributing to the development of Alzheimer’s disease and Creutzfeldt-Jakob disease and, as well as, at a cardiac level, cardiac amyloidosis [30]. Cardiac amyloidosis is characterized by accumulation of extracellular misfolded proteins; nine proteins have been identified as being able to aggregate as amyloids [31]. Amongst these proteins, TTR—in both hereditary and acquired forms—is the main protein involved in cardiac amyloidosis. Initially, cardiac amyloidosis was considered to be a rare disease of elderly patients; however, the advances in techniques for diagnosis and postmortem studies have revealed that it was underappreciated as a cause of common cardiac diseases or syndromes [32], and it has been reported as a frequent cause of heart failure with preserved ejection fraction and severe aortic stenosis [33]. The survival of patients after diagnosis varies, depending on the type of amyloidosis, with a survival time of approximately 60 months in TTR-related amyloidosis [31]. The clinical manifestations of cardiac amyloidosis are restrictive cardiomyopathy with restrictive ventricular filling, cardiac hypertrophy, conduction abnormalities and atrial arrythmias [34]. Until now, the standard treatment for cardiac amyloidosis has been the use of tafamidis which has exhibited a lower all-cause mortality in patients with cardiac amyloidosis; however, its use has several limitations, such as the excessive price of treatment and the need for a high frequency of dosing [35]. For all of these reasons, it is necessary to deepen the understanding of the basis of the pathology in order to discover new mechanisms and possible treatments.

A link between TTR and ER stress has been proposed. TTR is dissociated into monomers due to ER stress pathways, which can facilitate destabilization of TTR. In our study, obese animals presented with cardiac ER stress [30]. These data are in agreement with the observation in the animal model, in which obese animals present with cardiac ER stress. In addition, treatment with the mitochondrial antioxidant MitoQ not only prevented the activation of ER stress but increased TTR plasma and cardiac levels in obese animals. In addition, TTR levels were correlated with cardiac fibrosis and collagen type I protein levels, showing a possible relationship between TTR and the development of cardiac fibrosis, as well as an interaction with mitochondrial oxidative stress.

Cardiac fibrosis is a common feature observed in obesity, which ultimately leads to myocardial stiffness, arrhythmias and diastolic dysfunction [10,36]. Previous studies have confirmed the detrimental role of TTR in cardiac cells, showing toxic effects on cardiac myocytes [37]; however, there are no studies evaluating the effects of TTR on extracellular matrix synthesis. We showed, for the first time, the role of TTR in the development of cardiac fibrosis in cardiac fibroblasts, the main cell type involved in collagen synthesis in the heart [38]. In the cells, TTR was able to promote collagen synthesis in a dose-dependent manner, as well as an increase in BiP protein levels, thus showing ER stress activation. In cardiac fibroblasts, TTR was also able to increase CHOP protein levels, a downstream target upregulated under ER stress. CHOP is a common downstream element of inositol-requiring 1 and PKR-like ER kinase, which is involved in cardiac apoptosis, hypertrophy and heart failure [39]. Complementary analyses show that TTR only modified ATF6α protein levels in cardiac cells at the highest dose of 10 μg/mL. Previous studies have demonstrated the role of ATF6α in cardiac alterations promoting cardiac hypertrophy, as its specific deletion in cardiac myocytes blunted the development of cardiac hypertrophy and impaired cardiac function in an animal model of transverse aortic constriction [40]. However, the role of ATF6α in cardiac fibrosis remains controversial, as some studies have shown that ATF6α decreases the differentiation of cardiac fibroblasts in myofibroblasts, reducing cardiac fibrosis [41]. In contrast, ATF6α was upregulated by the profibrotic mediator TGF-β in human cardiac fibroblasts, thus contributing to the development of cardiac fibrosis [42]. All these data suggest that TTR exerts profibrotic actions and ER stress activation through its pro-oxidant actions. This affirmation is based on treatment with the mitochondrial antioxidant MitoQ, which prevented all these effects of TTR. Therefore, mitochondrial oxidative stress not only mediates the increase in TTR protein levels observed in obese animals but also mediates the cardiac actions of TTR.

In summary, through proteomic analyses, we identified an increase in TTR protein levels in obese animals, an effect that was mediated by mitochondrial oxidative stress. The protein interactome network showed that TTR is related to ER stress, a pathological mechanism involved in the development of cardiac fibrosis. In vitro studies revealed that TTR promoted an increase in collagen type I protein levels and ER stress activation through mitochondrial oxidative stress. The results revealed another mechanism involved in cardiac fibrosis associated with obesity and the beneficial effects of treatment with mitochondrial antioxidants.

Some limitations should be addressed. This study was performed only in male rats to avoid the effect of modulation of estrogen on the cardiovascular system; however, a comparison between the two sexes could provide a more comprehensive view of how obesity and mitochondrial oxidative stress could impact transthyretin levels and effects. In addition, proteomics were performed in bulk tissue. Future deployment of single-cell approaches will allow us to define in detail the localization, expression and role of TTR in each cardiac cell types, increasing our knowledge about the molecular mechanisms involved in the cardiac alterations associated with obesity. This study highlights that mitochondrial oxidative stress mediates TTR effects on cardiac fibroblasts; however, an appropriate animal model for TTR-derived cardiac amyloidosis treated with MitoQ could show the possible beneficial effects of mitochondrial antioxidants in the pathology.

## 4. Materials and Methods

### 4.1. Animal Model

Six-week-old male Wistar rats with an initial body weight of 150 g (Envigo, Barcelona, Spain) were fed a standard diet ad libitum (CT, 3.5% fat; Envigo Teklad no.TD.2014, Haslett, MI, USA; n = 16) or a high-fat diet (HFD, 35% fat; Envigo Teklad no. TD.03307, Haslett, MI, USA; n = 16) for 7 weeks. Diet compositions are presented in Appendix A. Half of the animals in each group received the mitochondrial antioxidant MitoQ (200 μM) in their drinking water, as previously described [16]. The Animal Care and Use Committee of Universidad Complutense de Madrid approved all experimental procedures according to the Spanish Policy for Animal Protection RD53/2013, which meets European Union Directive 2010/63/UE.

### 4.2. Cell Culture Studies

Cardiac fibroblasts from Innoprot (Ref: P10402) were used between passages 7 and 8 and were cultured under the same conditions (37 °C, 95% sterile air and 5% CO_2_) in a saturation humidified incubator (Heracell 150i, Heraeus, Germany) following the manufacturer’s instructions. Cells were maintained in DMEM supplemented with 10% fetal bovine serum, 10 mmol/L L-glutamine, 100 U/mL penicillin/streptomycin, 10 mmol/L L-pyruvate and 2 mmol/L HEPES. Cells were seeded into six-well plates at 90% confluence and serum-starved for 18–24 h. Cells were treated for 24 h with different doses of recombinant TTR (0.5–10 μg/mL; Abexxa; Ref: abc069456) in the presence or absence of MitoQ (5 nM). The doses of the treatments were based on previous studies [16,43]. MitoQ was provided by MP Murphy from the Medical Research Council Mitochondrial Biology Unit, Cambridge BioMedical Campus, Cambridge, UK.

### 4.3. Proteomic Analyses

#### 4.3.1. Sample Preparation

Tissue samples were homogenized in lysis buffer (7 M urea, 2 M thiourea, 50 mM DTT), and protein concentration was quantified with a Bradford assay kit (Bio-Rad) and precipitated with a ReadyPrep 2-D cleanup kit (BioRad). The protein extract for each sample was diluted in Laemmli buffer and loaded into a 1.5 mm thick polyacrylamide gel with a 4% stacking gel casted over a 15% resolving gel. The run was stopped as soon as the front entered 3 mm into the resolving gel to concentrate the whole proteome in the stacking/resolving gel interface. Bands were stained with Coomassie Brilliant Blue and excised from the gel. Purification and concentration of peptides were performed using C18 Zip Tip solid-phase extraction (Millipore).

#### 4.3.2. Mass Spectrometry Analysis

Peptide mixtures were separated by reverse-phase chromatography using an Eksigent nanoLC ultra 2D pump fitted with a 75 μm ID column (Eksigent 0.075 × 250). Samples were first loaded into a 0.5 cm length, 100 μm ID precolumn packed with the same chemistry as the separating column for desalting and concentration. Mobile phases were 100% water, 0.1% formic acid (buffer A) and 100% acetonitrile 0.1% formic acid (buffer B). Column gradient was developed in a 200 min, two-step gradient from 5% B to 25% B in 160 min and 25%B to 40% B in 21 min. The column was equilibrated in 95% B for 8 min and 5% B for 11 min. During all processes, the precolumn was in line with the column, and the flow was maintained along the gradient at 300 nL/min. Eluting peptides from the column were analyzed using a Sciex 5600 Triple-TOF system. Data were acquired by survey scan performed in a mass range from 350 m/z up to 1250 m/z with a scan time of 250 ms. The top 35 peaks were selected for fragmentation. Minimum accumulation time for MS/MS was set to 100 ms, for a total cycle time of 3.8 s. Product ions were scanned in a mass range from 230 m/z up to 1500 m/z and excluded for further fragmentation for 15 s.

#### 4.3.3. Data Analysis

MS/MS data acquisition was performed using Analyst 1.7.1 (Sciex), and spectra files were processed through Protein Pilot Software (v.5.0.1-Sciex) using the Paragon™ algorithm (v.5.0.1) for database search [44] and Progroup™ for data grouping and searched against the concatenated target–decoy UniProt proteome database (rat norvegicus). The false-discovery rate was determined using a non-lineal fitting method [45] and results reporting a 1% global false discovery rate or better were displayed.

Peptide quantification was performed using Progenesis LC−MS software (2.0.5556.29015, Nonlinear Dynamics). Using the accurate mass measurements from full survey scans in the TOF detector and the observed retention times, runs were aligned to compensate for between-run variations in our nanoLC separation system. To this end, all runs were aligned to a reference run automatically chosen by the software, and a master list of features considering m/z values and retention times was generated. The quality of these alignments was manually supervised with the help of quality scores provided by the software. The peptide identifications were exported from Protein Pilot software and imported to Progenesis LC−MS software, where they were matched to the respective features. Output data files were managed for subsequent statistical analyses and representation. Proteins identified by site (identification based only on a modification), reverse proteins (identified by decoy database) and potential contaminants were filtered out. Proteins quantified with at least two unique peptides, *p*-value lower than 0.05 and an absolute fold change of <0.7 or >1.3 in linear scale were considered significantly differentially expressed.

Bioinformatic analysis was performed using Metascape and Biogrid 4.4 tools [46,47] with default settings.

### 4.4. Western Blot

For Western blog analysis, 20 μg of total cardiac proteins or cell lysates was separated by sodium dodecyl sulfate–polyacrylamide gradient gel (4–20%; BioRad, Hercules, CA, USA) and transferred to Hybond-c Extra nitrocellulose membranes (Hybond-P; Amersham Biosciences, Piscataway, NJ, USA) with the Trans-Blot Turbo Transfer System. Membranes were probed with primary antibody for collagen type I (Calbiochem, San Diego, CA, USA; dilution 1:500; Ref: 234167), BiP (BD Biosciences, Madrid, Spain; dilution 1:1000; Ref: 610978), CHOP (Cell Signaling Technology, Danvers, MA, USA; dilution 1:500; Ref: #2895), ATF6α (Santa Cruz, Dallas, TX, USA; dilution: 1:250; Ref: sc-166659) and glyceraldehyde-3-phosphate dehydrogenase (GAPDH; Cell Signaling; dilution: 1:5000; Ref: #5174) as loading control. The signals were detected using an ECL system (Millipore, Burlington, MA, USA). Several proteins were analyzed in the same membrane after a stripping procedure (Thermo Scientific, Waltham, MA, USA; Ref: 21063). The results are expressed as n-fold increases over the values of the control group in arbitrary densitometric units.

### 4.5. Circulating Plasma Levels of TTR

Circulating TTR levels were measured by sensitive enzyme immunoassays (Abnova; Ref: KA2137) following the instructions of the manufacturer.

### 4.6. Statistical Analyses

Data are expressed as mean ± SEM. Normality of distributions was verified by means of the Kolmogorov–Smirnov test. Pearson correlation analysis was used to examine association among different variables. Data were analyzed using a one-way analysis of variance, followed by a Tukey test or Dunnett test to assess specific differences among doses or control conditions, respectively, using GraphPad software (San Diego, CA, USA). The predetermined significance level was α equal to 0.05.

## Figures and Tables

**Figure 1 ijms-23-08080-f001:**
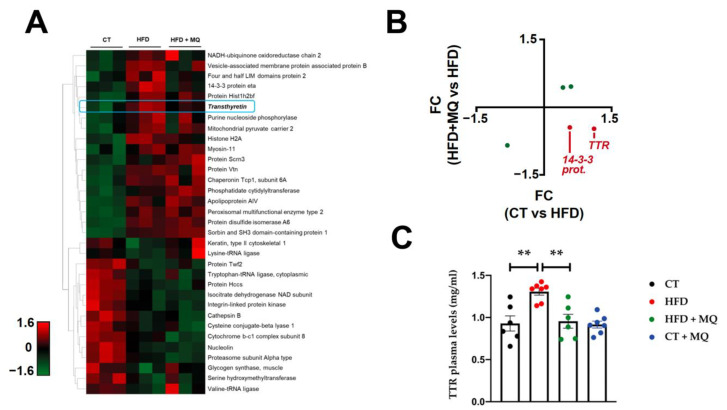
Mitochondrial oxidative stress promotes transthyretin upregulation in obesity. Heat map representation showing proteins differentially expressed and the degree of change in hearts from rats fed a standard diet (control, CT) or a high-fat diet (HFD) treated with MitoQ (HFD + MQ) (**A**). Correlations in cardiac samples between (only significant) protein modulations (FC) under HFD and HFD + MQ conditions (red dots represent proteins with reversal of changes observed in obese rats after MQ treatment) (**B**). Circulating plasma levels of transthyretin (TTR) of rats fed a standard diet (control, CT) or a high-fat diet (HFD) treated with MitoQ (HFD + MQ) or vehicle (CT + MQ) (**C**). Bar graphs represent the mean ± SEM of 6–8 animals. ** *p* < 0.01.

**Figure 2 ijms-23-08080-f002:**
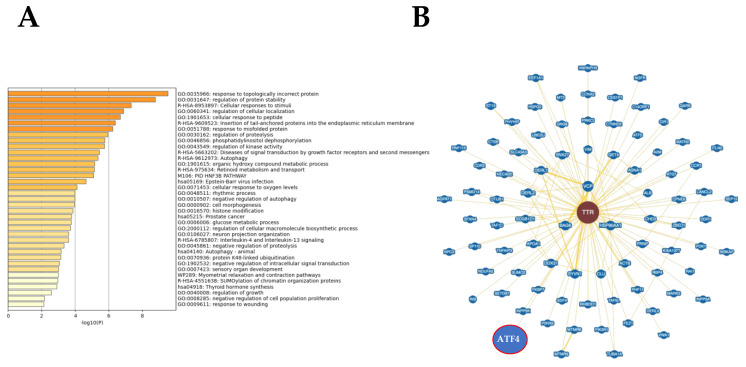
Transthyretin is associated with endoplasmic reticulum stress. Significantly enriched biofunctions detected through Metascape tool (**A**) and experimentally-demonstrated protein interactome network for transthyretin generated by the Biogrid database (**B**).

**Figure 3 ijms-23-08080-f003:**
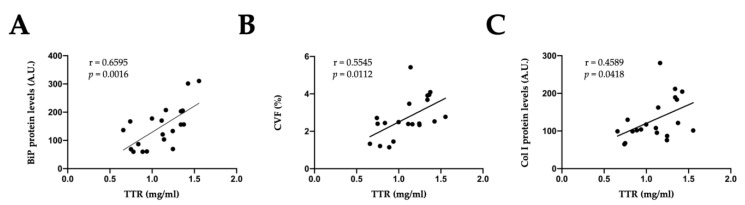
Circulating transthyretin is associated with endoplasmic reticulum stress and cardiac fibrosis in obese rats. Direct correlations between transthyretin (TTR) plasma levels and (**A**) binding immunoglobulin protein (BiP), (**B**) cardiac collagen volume fraction (CVF) and (**C**) collagen I (Col I) protein levels in all animals. r: Pearson’s correlation coefficient; A.U.: arbitrary units.

**Figure 4 ijms-23-08080-f004:**
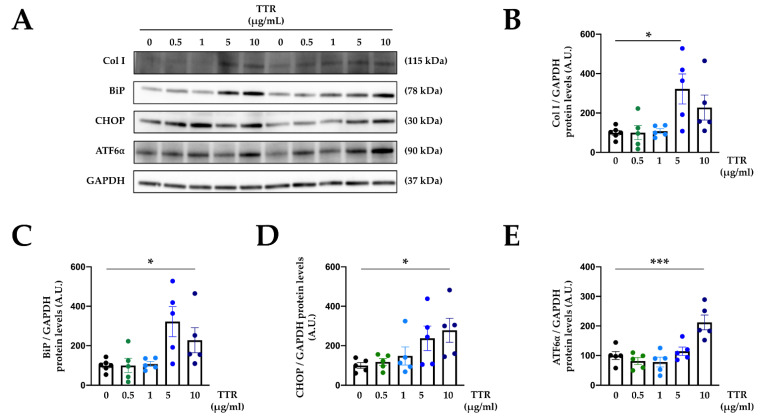
Transthyretin induces collagen I and endoplasmic reticulum stress activation in cardiac fibroblasts. Representative Western blots (**A**) and quantification of collagen I (Col I) (**B**), binding immunoglobulin protein (BiP) (**C**), CCAAT-enhancer-binding homologous protein (CHOP) (**D**) and activating transcription factor 6 (ATF6α) (**E**) in cardiac fibroblasts treated with transthyretin (TTR) for 24 h at different doses (0.5–10 μg/mL). Bar graphs represent the mean ± SEM of four to six assays normalized to glyceraldehyde-3-phosphate dehydrogenase (GAPDH). * *p* < 0.05; *** *p* < 0.001. A.U: arbitrary units.

**Figure 5 ijms-23-08080-f005:**
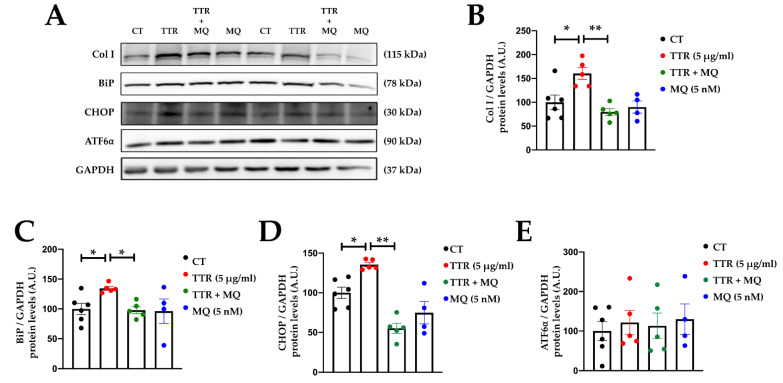
Mitochondrial oxidative stress mediates profibrotic and endoplasmic reticulum stress activation induced by transthyretin. Representative Western blots (**A**) and quantification of collagen I (Col I) (**B**), binding immunoglobulin protein (BiP) (**C**), CCAAT-enhancer-binding homologous protein (CHOP) (**D**) and activating transcription factor 6 (ATF6α) (**E**) in cardiac fibroblasts treated with transthyretin (TTR; 5 μg/mL) in the presence or absence of MitoQ (MQ; 5 nM) for 24 h. Bar graphs represent the mean ± SEM of four to six assays normalized to glyceraldehyde-3-phosphate dehydrogenase (GAPDH). * *p* < 0.05; ** *p* < 0.01. A.U: arbitrary units.

## Data Availability

MS data and search results files were deposited in the ProteomeXchange Consortium via the JPOST partner repository (https://repository.jpostdb.org, accessed on 29 June 2022) [48] with the identifier PXD034955 for ProteomeXchange and JPST001678 for jPOST (for reviewers: https://repository.jpostdb.org/preview/183811480262bc244894e7a, accessed on 29 June 2022; Access key: 2063).

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
