# Peer review of "Mitochondrial Oxidative Stress Induces Cardiac Fibrosis in Obese Rats through Modulation of Transthyretin"

_ijms, 2022, doi:10.3390/ijms23158080_

Round 1
Reviewer 1 Report
The authors discovered by proteomic analyses an increase in transthyretin protein levels (and 32 other proteins) in obese rats. This effect that was mediated by mitochondrial oxidative stress. Protein interactome network showed that transthyretin is related with endoplasmic reticulum stress, a pathological mechanism involved in the development of cardiac fibrosis.In general, this original research paper is well written and contains a coherent and interesting data set. It is important to put the model into perspective in the Discussion. Minor comments: 1. High-fat diet is a generic term. Please provide details of the composition of the specific diet. 2. Abstract: 'MitoQ modulates the expression of 23 proteins altered in obese animals, including the transthyretin (TTR), whose levels were higher in obese animals as compared to control animals and reduced in MitoQ-treated animals’. I understand what the authors mean but the whole sentence is rather awkward. Please rephrase. 3. Cardiac fibrosis is only one feature of pathological hypertrophy. Do the authors have other data in relation to pathological hypertrophy or data in relation to cardiac function? Do these rats develop diabetes? Are these rats in heart failure? According to info in lines 90-92, it is a model of rather mild cardiac changes. Please put this model into perspective. Detailed data on cardiac structure and function in a model of diabetic cardiomyopathy induced by a high-sugar high-fat diet can be retrieved in this IJMS paper: Int J Mol Sci. 2019 Mar 13;20(6):1273. doi: 10.3390/ijms20061273. PMID: 30871282 4. The authors state: 'Obesity is defined as abnormal or excessive fat accumulation and is considered a major risk to health’. This is not accurate. Obesity is defined based on body mass index and is not exactly the same as adiposity. Please make this distinction in the Introduction. 5. In general, there are too many non-standard abbreviations. Abbreviations reduce readability.
Reviewer 2 Report
This manuscript evaluates the potential of the antioxidant, MitoQ, to reverse cardiac fibrosis in obese rats. MitoQ administration, in vitro and in vivo. reversed the obesity-induced upregulation of transthyretin (TRR), in association with additional proteins related to ER stress. While this manuscript is well-written, authors need to further describe their methods, especially in vivo, and address a potential issue with figure 5.
Major:
1. In Methods, Lines 281-289, were the rats fed ad libitum? If so, how can authors be sure rats ate same amount? Was weight gain/loss compared between groups? Were the HF and standard diet isocaloric? Same protein percentage? Why were only male rats used? The rats were all the same weight? What were their ages?
What media were the fibroblasts kept in?
2. In Figure 5, do authors have the quantification data for GAPDH in the Western blot? Looks like the quantity varies among samples, which would bring into question other differences shown in that Figure.
3. Please include a small discussion of the limitations of this study at the end of the discussion.
Minor:
1. In Intro, Line 43, delete first instance of “increases”.
2. Line 50—write out morbidity and mortality, rather than hyphenating as “morbi-mortality”.
3. Lines 64-81—this portion of the paragraph is confusing. Please write more clearly.
4. Line 343, use decimal instead of comma for numbers, as was done throughout remainder of manuscript.
5. Line 234, misspelled dosing.
Round 2
Reviewer 2 Report
Thanks for the edits.
1. The information on the diets and weight changes is important, and it wasn’t clear that this information, in particular, had already been evaluated in a previous study. Can the authors either republish the body weight graph/food and energy intake table in supplementary material, or note that that information exists somewhere in the Results (citing their previous study where that information was published)? The information on why only male rats were used also seems important enough to include—can authors simply add a small phrase to their limitations paragraph to satisfy this?
2. In limitations paragraph, “proteomic was” should be “proteomics were” (Line 301).
